
# Temporal changes in rainfall intensity-duration thresholds for post-
# wildfire flash floods and sensitivity to spatiotemporal distributions of
# rainfall
Tao Liu[1,2], Luke A. McGuire[1], Nina Oakley[3], Forest Cannon[3]
[1]Department of Geosciences, University of Arizona, Tucson, AZ 85721-0011, USA
[2]Department of Hydrology and Atmospheric Sciences, University of Arizona, Tucson, AZ 85721-0011, USA
[3]Center for Western Weather and Water Extremes, Scripps Institution of Oceanography, University of California, San Diego,
La Jolla, CA, USA
*Correspondence to*: Tao Liu (liutao@arizona.edu)
**Abstract.** Rainfall intensity-duration (ID) thresholds are commonly used to assess flash flood potential downstream of burned
watersheds. High-intensity and/or long-duration rainfall is required to generate flash floods as landscapes recover from fire,
but there is little guidance on how thresholds change as a function of time since burning. Here, we force a hydrologic model
with radar-derived precipitation to estimate ID thresholds for post-fire flash floods in a 41.5 km$^2$ watershed in southern
California, USA. Prior work in this study area constrains temporal changes in hydrologic model parameters, allowing us to
estimate temporal changes in ID thresholds. Results indicate that ID thresholds increase by more than a factor of 2 from post-
fire year 1 to post-fire year 5. Thresholds based on averaging rainfall intensity over durations of 30-60 minutes perform better
than those that average rainfall intensity over shorter time intervals. Moreover, thresholds based on the 75$^{th}$ percentile of radar-
derived rainfall intensity over the watershed perform better than thresholds based on the 25$^{th}$ or 50$^{th}$ percentile of rainfall
intensity. Results demonstrate how hydrologic models can be used to estimate changes in ID thresholds following disturbance
and provide guidance on the rainfall metrics that are best suited for predicting post-fire flash floods.



## 1 Introduction

Heightened hydrologic responses are common within and downstream of recently burned areas, resulting in an increased likelihood of flash floods. Rainfall intensity-duration (ID) thresholds are commonly used to assess the potential for flash floods (Moody and Martin, 2001; Cannon et al., 2008). Many past studies aimed at defining thresholds for flash floods focus on the first 1-2 years following fire (Cannon et al., 2008; Wilson et al., 2018). Since the hydrologic impacts of fire are transient, rainfall ID thresholds associated with flash floods are likely to change as a watershed recovers (Ebel and Martin, 2017; Ebel and Moody, 2017; Moreno et al, 2019; Ebel, 2020). It may take more than a decade for hydrologic responses to return to pre-fire levels, yet there is limited guidance on how the magnitude and utility of rainfall ID thresholds change with time since burning. Given the increased frequency and size of fire in many geographic and ecological zones (e.g. Gillett et al., 2004; Westerling et al., 2006; Kitzberger et al., 2017), it is of growing importance to quantify the best metrics for assessing flash-flood potential in the immediate aftermath of fire as well as how these metrics change throughout the recovery process (e.g. Ebel, 2020).

Rainfall ID thresholds for flash floods are typically defined using historic data that relates rainfall over different intensities and durations to an observed hydrologic response, namely the presence or absence of flooding (e.g. Cannon et al., 2008). Due to the stochastic nature of rainfall over burned areas and limited observations throughout the recovery process, there is a paucity of data that can be used to derive empirical thresholds for flash flooding beyond one year of recovery. Hazards associated with flash flooding, however, may exist downstream of burned areas well beyond one year of recovery. Wildfire alters rainfall-runoff partitioning and flood routing by incinerating vegetation and reducing interception capacity (Stoof et al., 2012, Saksa et al., 2020), decreasing hydraulic roughness, and reducing soil infiltration capacity (Larsen et al., 2009, Ebel and Moody, 2013). Reductions in infiltration capacity are often attributed to fire-induced soil water repellency (Ebel and Moody, 2013), which is generally strongest immediately following a fire and then decays over time scales ranging from one year to more than five years (Dyrness, 1976; Huffman et al., 2001; Larsen et al., 2009), though surface soil sealing (Larsen et al., 2009) and hyper-dry conditions (Moody and Ebel, 2012) are also known to play important roles. Vegetation recovery, which may influence temporal changes in hydraulic roughness and canopy interception, can take five years or longer. Cannon et al. (2008) collected sufficient data over a two-year time period following fire in southern California, USA, to define separate rainfall ID thresholds for post-fire debris flows and flash floods in the first- and second-years following fire. They found that the ID thresholds for flash floods and debris flows may increase by as much as 25 mm/h after one year of recovery, a change that they attributed to a combination of vegetation growth and sediment removal as a result of rainstorms during the first post-fire year.

Rainfall ID thresholds are often defined over a range of durations, though averaging rainfall intensity over a particular duration may provide a more reliable threshold. Post-fire hydrological response in the first few years is often best related to rainfall



intensity over short durations (less than 60 min) (Staley et al., 2017; Moody and Martin, 2001). In their efforts to define rainfall

ID thresholds for post-fire debris flows, Staley et al. (2013) showed that averaging rainfall intensities over durations between

15 minutes and 60 minutes resulted in thresholds that performed better relative to those associated with longer durations. One

potential explanation for this observation is that post-fire debris flows are often triggered by runoff in steep, low-order

drainages, which both Kean et al. (2011) and Raymond et al. (2020) have found to be highly correlated with rainfall intensities

averaged over similarly short time intervals (10-15 minutes). Moody and Martin (2001) have also documented a substantial

increase in peak discharge following wildfire once the 30-minute rainfall intensity ($I_{30}$) crossed a threshold value, suggesting

that $I_{30}$ may be a consistent predictor of flash flood activity in recently burned watersheds. Moody and Martin (2001) suggest

that peak $I_{30}$ can be used to set the threshold for early-warning flood systems. The optimal duration for defining post-fire flash

floods thresholds, as well as how it may change with time, remains relatively unexplored.

Rain gage records are typically used to derive rainfall ID thresholds for flash flood and post-fires debris flows (Staley et al.,

2013; Staley et al., 2017). Post-fire debris flows, however, tend to initiate in small (<1 km$^2$), steep watersheds. In these small

watersheds, the rainfall intensity responsible for initiating a debris flow can be characterized by a single rain gage installed

near the initiation zone. Flash floods differ in that they tend to occur at larger spatial scales where rainfall is spatially variable

and may not be adequately characterized by data from a single rain gage. Radar-derived precipitation estimates, which can

provide high spatiotemporal resolution of rainfall intensity, present opportunities to develop basin-specific thresholds for post-

fire flash floods. However, high spatiotemporal variability in rainfall intensity also brings new challenges when employing

radar-derived precipitation in flood warning practice. In particular, what is the best way to summarize spatially and temporally

variable rainfall intensity information with a single metric that can be used as a threshold? How does hydrological recovery

following fire influence the generation of flash floods and the metrics that are best suited for their prediction? Data-driven

approaches to answering these and related questions may be hampered by limited monitoring of post-fire hydrologic response

throughout the recovery period and the stochastic occurrence of rainfall over burned areas, which limits opportunities for

observations. Given a well-constrained hydrologic model that accounts for changes associated with post-fire recovery, it is

possible to use numerical experiments to understand relationships between time since burning, the spatiotemporal patterns of

rainfall over a watershed, and the occurrence of flash floods.

Here, we use realistic patterns of spatially and temporally varying radar-derived rainfall over a 41.5 km$^2$ watershed in the San

Gabriel Mountains of southern California, USA, to (1) determine the optimal method to define a rainfall ID threshold for flash

floods, and (2) identify changes in rainfall ID thresholds for flash floods as a function of time since burning. The watershed,

which we refer to as the upper Arroyo Seco, burned during the 2009 Station Fire (USDA Forest Service, 2009). Liu et al.

(2021) used rain and stream gage data collected at different times following the fire to calibrate the KINEROS2 hydrologic

model for this watershed, enabling them to quantify temporal changes in model parameters as a function of time since burning.

Combining this calibrated model with spatially explicit, radar-derived estimates of rainfall intensity during 34 rainstorms, we



explore the utility of different rainfall ID metrics as flash flood thresholds and quantify temporal changes in those thresholds
through the first five years of recovery. Results provide insight into the magnitude of temporal changes in flash flood thresholds
in the densely populated, fire-prone region of southern California. More generally, results support the development of early
warning systems for flash floods by identifying specific metrics that can be computed using spatially variable rainfall intensity
estimates to assess the potential for flash flooding.
**2 Study Area**

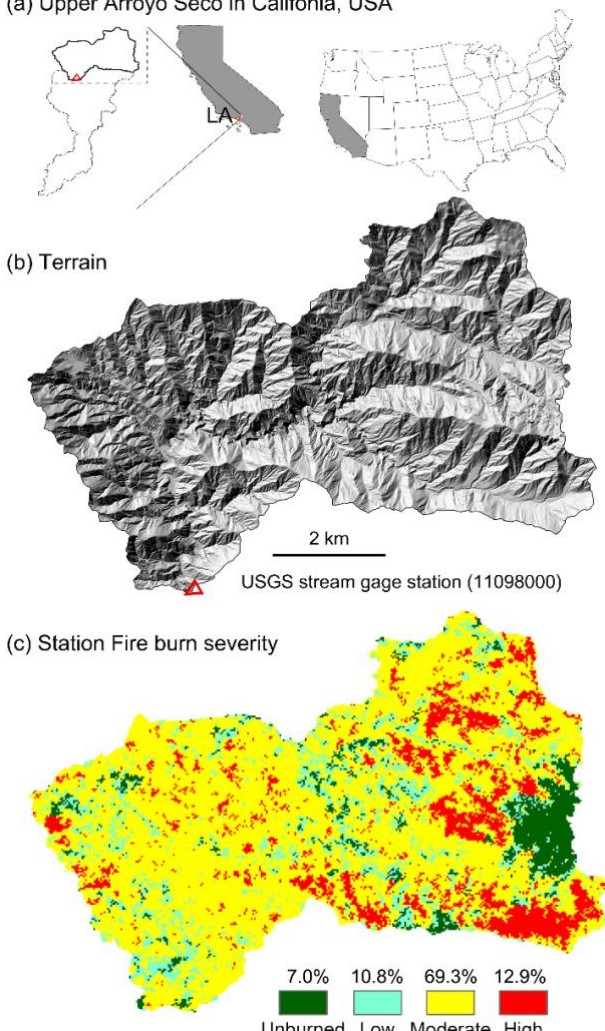


**Figure 1: Modified from figure 1 in Liu et al. (2021) (a) The location of the upper Arroyo Seco watershed within California. The red**
**triangle indicates the location of the USGS stream gage (11098000); (b) Shaded relief showing the study watershed with the USGS**



**stream gage (red triangle; 34°13'20", -118°10'36"); (c) Soil burn severity for the 2009 Station fire. Burn severity percentages are for**
**planform area within each category.**

The upper Arroyo Seco watershed drains the 41.5 km$^2$ area above USGS stream gage station (11098000) near Pasadena in the
San Gabriel Mountains (Figure. 1). The upper Arroyo Seco was burned in the August-October 2009 Station Fire, which burned
more than 80% of the watershed at moderate to high soil burn severity (USDA Forest Service, 2009). Dominant shrubs and
chaparral, such as chamise (*Adenostoma fasciculatum*) and manzanita (*Arctostaphylos spp.*), were completely consumed with
severe soil heating in isolated patches throughout many areas burned at moderate to high severity (USDA Forest Service,
2009). Soils in this area are typically sand and silty-sand textured and thin (<1 m) with partial exposure of bedrock (Staley et
al., 2014). The majority of rainfall in the study area typically occurs in the cool season, between December and March, while
warm, dry conditions dominate from April to early November. The San Gabriel Mountains also experience some of the most
frequent short-duration, high-intensity rainfall in the state (Oakley et al. 2018a).

Due to wildfire-induced changes in surface conditions, including canopy cover and soil-hydraulic properties, runoff generation
in the first year following the fire was likely dominated by infiltration excess overland flow (Schmidt et al., 2011, Liu et al.,
2021). Enhanced soil water repellency (SWR), which helps promote low infiltration capacity, and extensive dry ravel, which
loads channels with fine-grained hillslope sediment, are both commonly observed after fires in the San Gabriel Mountains
(e.g., Watson and Letey, 1970; Hubbert and Oriol, 2005; Lamb et al., 2011; Hubbert et al., 2012). Rengers et al. (2019)
calibrated a hydrologic model using data from small watersheds (0.01-2 km$^2$) burned by the Station Fire and found relatively
low values for saturated hydraulic conductivity ($K_s$), generally between 2-10 mm/h. These results are consistent with values
for saturated hydraulic conductivity inferred by Liu et al. (2021) via model calibration in the upper Arroyo Seco watershed.
The impact of dry ravel, which reduces grain roughness in the channel network, and reduced vegetation density led to estimates
of Manning's $n$ in the channels of the upper Arroyo Seco of approximately 0.09 s m$^{-1/3}$ in the first year following fire (Liu et
al., 2021). These hydrologic changes led to widespread flooding and debris flows during multiple rainstorms in the first winter
after the fire (Kean et al., 2011; Oakley et al., 2017). As hydrologic recovery began over the next several years, the watershed-
scale $K_s$ and Manning's $n$ generally increased and likely started to mitigate the flash flood risk (Liu et al., 2021).
**3 Data and Methods**
**3.1 Radar-derived precipitation**
We sought to identify storms in the study area that produced moderate-to-high intensity rainfall to use as inputs to a hydrologic
model to simulate flood responses. Storm events were selected within the period for which observations are archived for the



two operational NWS Next-Generation Weather Radar installations (NEXRAD; NOAA 1991) that cover the study area,
KSOX, (Santa Ana), and KVTX (Ventura). Though archives for the radars begin in 1997 and 1995, respectively.

We compiled storm events starting with those known to have produced high intensity rainfall and a debris flow response in
the San Gabriel Mountains (e.g., Table 1 in Oakley et al. 2017) as well as other storms that produced high-intensity rainfall in
the region (e.g., Oakley et al. 2018b, Cannon et al. 2018). We then used hourly rainfall observations from the Clear Creek
(2002-present), San Rafael Hills (2005-present), and Heninger Flats (2010-present) Remote Automated Weather Stations
(RAWS, acquired from raws.dri.edu) as indicator gages for the study area. This further limited us to post-2002 events outside
of the literature. All gages are <10 km from the watershed of interest; there were no long-record gages within the watershed.
We used 15 mm/h as a threshold for moderate to high intensity rainfall and extracted all events from the gauge record meeting
or exceeding this value to develop a list of events of interest. We reviewed the radar data for these events at which point some
of the selected events could not be utilized due to radar outages or poor data quality. This exercise presented us with 34 storm
events (Table S1).

Various atmospheric processes may contribute to generation of moderate-to-high rainfall intensities (e.g., Oakley et al. 2017),
resulting in differing spatial and temporal precipitation patterns over a burn area. To ensure the events selected captured
variability in spatial and temporal precipitation characteristics, we evaluated the spatial characteristics of the events. We found
rainfall patterns could generally be categorized into four main spatial patterns at the scale of several tens of kilometers: (1) a
broad pattern, a contiguous area of moderate-to-high intensity precipitation (>45 dBZ) spanning tens of kilometers; (2) a
scattered pattern with numerous cells of moderate to high precipitation that are not spatially continuous; (3) an isolated pattern,
with one to a few isolated cells of moderate-to-high intensity rainfall separated by non-precipitating areas several to tens of
kilometers in extent; (4) a narrow cold frontal rainband (NCFR)—a north-south oriented narrow band (~3-5 km wide, tens to
100 km in length) of very high intensity rainfall (e.g., Oakley et al. 2018b; Cannon et al. 2020; Figure S1 in Supplement). At
the <10 km horizontal scale (the scale of the watershed), it was harder to identify meaningful patterns and distinctions, though
the larger scale signals imply varying spatial and temporal patterns of precipitation as each pass over the watershed. A table
of storm events and their characteristics is available in Table S1 in the Supplement.

An approximate start and end time were determined for each event using the Clear Creek RAWS gauge as an indicator. Start
time was determined by identifying the time of maximum 1h rainfall in the event and going back in time to the first of three
consecutive hours of >1.5 mm/h precipitation. The end of an event was determined as the last hour where precipitation dropped
below 3 mm/h for at least two consecutive hours.


Level-II base reflectivity (https://www.ncdc.noaa.gov/wct/) between the start and end time of each event was downloaded
from both the KSOX and KVTX radars. The data were used to generate spatially-distributed precipitation over the study area.
Radar imagery concurrent with the gauge-based record of high intensity rainfall events was converted to a composite maximum
reflectivity product at 250 m spatial and 5-minute temporal resolution. Conversion of radar reflectivity to rain rate required
the application of an empirically derived reflectivity (Z) to rain rate (R) relationship (e.g. Marshall and Palmer 1948). The Z-
R relationship is conventionally represented by the equation $Z = aR^b$, which includes parameters a and b to account for
variations in precipitation for a given reflectivity arising from differences in the drop size distribution. Due to the lack of
previous studies investigating Z-R relationships in precipitating conditions over the region of interest, there are no standard a
and b parameters to apply to the reflectivity data analyzed here. Thus, five well-known and previously published Z-R
relationships were applied to the gridded reflectivity values. Supplement S3 lists the different Z-R relationships applied here
and the general conditions for which they are suitable. Although the Z-R relationships used here are not based on observations
from the present study's region of interest, the variation of a and b parameters yields an estimate of precipitation uncertainty.
It is worth noting that a number of additional sources of radar measurement uncertainty exist that are not evaluated in depth
here, including beam broadening, topographic blocking and scan elevation. However, this was not of primary concern since
the goal of this study was to generate realistic spatial and temporal patterns of rainfall over the watershed with varying intensity
that could be used to force the KINEROS2 hydrologic model. The goal was not to reproduce the observed hydrologic response
resulting from a particular set of rainstorms.

As a range of precipitation intensities for each storm result from the application of the five different Z-R relationships (e.g.,
Figure S2 in Supplement), we utilize these as realistic storms of varying precipitation intensity to increase our storm sample
size, such that we apply 34 storms * 5 Z-R relations = 170 precipitation scenarios as inputs to KINEROS2. These 170 scenarios
were then processed for ingestion into KINEROS2 (Figure. 2).



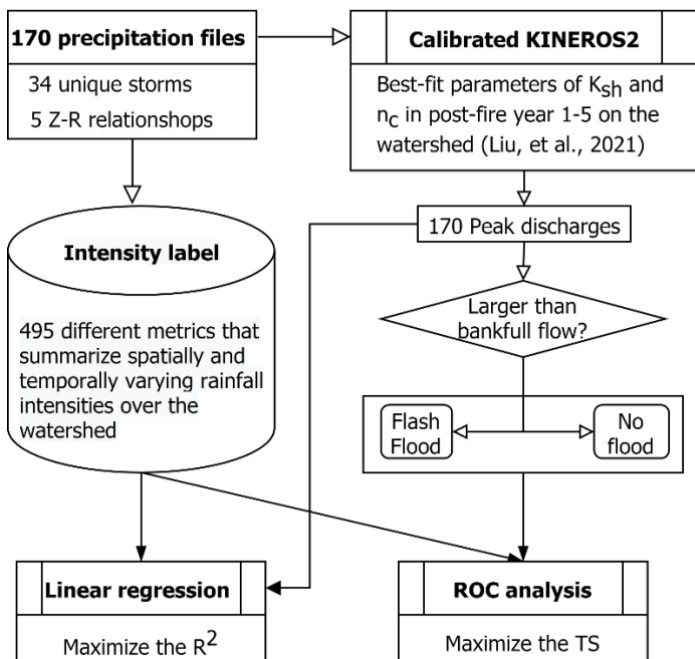


**Figure 2: Delineation of rainfall intensity-duration threshold for post-fire flash flood**

**3.2 Summary metrics for spatially and temporally varying rainfall**
In search of a spatiotemporel summary metric that may serve as a reliable flash flood threshold, we begin by describing a
methodology to summarize spatially and temporally varying rainfall over a watershed. For a given rainstorm, the rainfall
intensity time series at a single point, such as a single radar pixel, can be summarized by computing a moving average of
intensity over a specified duration, $D$. Letting $t$ denote time and $R$ denote the cumulative rainfall (mm), we define the rainfall
intensity over a duration $D$ at any given pixel within the watershed as

$$I_D(t) = \frac{R(t) - R(t-D)}{D} \tag{1}$$


Here, we compute $I_D(t)$ for each pixel for durations of 5, 10, 15, 30, and 60 minutes. Since the intensity in each radar pixel
could have a unique value, we also need a way to summarize $I_D(t)$ in space. One option would be to take the median of $I_D(t)$
to determine a typical value of $I_D$ within the watershed at each time, $t$. However, the median may not be a good predictor of
flash flooding since one could envision a scenario where it is only raining over 1/3 of the watershed, yet it is raining with
sufficient intensity to generate a flash flood. We therefore compute the j$^{\text{th}}$ percentile of $I_D(t)$ at each time, $t$, for j between 1
and 99. We denote the j$^{\text{th}}$ percentile of $I_D(t)$ as $I_D^j(t)$. For each rainstorm, we focus our analysis on the peak value of $I_D^j(t)$


which we denote as $I_D^j$. As an example, $I_{30}^{50}$ would be computed by defining $I_{30}$ for all radar time steps within a rainstorm,
determining the median value of $I_{30}$ over the watershed at each of those time steps, and then taking the maximum of that time
series of median $I_{30}$ intensities. This analysis yields 495 different metrics ($I_D^j$ for $j$=1,2,…,99 and $D$=5,10,15,30,60) that
summarize spatially and temporally varying rainfall intensities over the watershed. In the following sections, we describe how
we test the utility of each of these 495 different metrics as a flash flood threshold.
**3.3 Hydrological modeling**
We used the KINEROS2 (K2) hydrological model to simulate the rainfall partitioning, overland flow generation, and flood
routing in the upper Arroyo Seco watershed. K2 is an event-scale, distributed-parameter, process-based watershed model,
which has been used extensively for rainfall-runoff processes in semi-arid and arid watersheds (Smith et al., 1995; Goodrich
et al., 2012). Liu et al (2021) used rain gage data in combination with the USGS stream gage installed at the outlet of the upper
Arroyo Seco watershed to calibrate K2 during different stages of the post-fire recovery process. We use the same model setup
for simulations in this study. In particular, the 41.5 km$^2$ watershed was discretized into 1289 hillslope planes and these planes
were connected by a stream network of 519 channel segments based on a one-meter LiDAR-derived digital elevation model
(DEM). After accounting for a fixed interception depth of 2.97 mm based on land cover look-up table in the Automated
Geospatial Watershed Assessment toolkit (AGWA; Miller et al., 2007), infiltration of rainfall into soil is represented using the
Parlange et al. (1982) approximation. Overland flow and channel flow are modeled by kinematic wave equations. Both
saturated hydraulic conductivity on hillslopes ($K_{sh}$) and hydraulic roughness in channels ($n_c$) primarily determine runoff
generation and the shape of hydrograph, including total runoff volume, peak discharge rate, time to peak (Canfield et al., 2005;
Yatheendradas et al., 2008; Menberu et al., 2019). Other parameters, such as hydraulic roughness ($n_h$) and capillary drive ($G_h$)
on hillslopes, had a relatively minor impact on modelled runoff after the Station Fire in the upper Arroyo Seco watershed (Liu
et al., 2021).

**Table 1. Summary of model parameters for post-fire year 1, 2, 3, and 5. The saturated hydraulic conductivity on**
**hillslopes ($K_{sh}$) and hydraulic roughness in channels ($n_c$) are the average of values calibrated in post-fire years 1, 2,**
**3, and 5 (Liu et al., 2021)**

| Post-fire Year | Calibration Events | $K_{sh}$ (mm/hr) | $n_c$ (s/[m$^{1/3}$]) |
|---|---|---|---|
| 1 | 12 Dec 2009 | | |
| | 17 Jan 2010 | 7.2 | 0.087 |
| | 5 Feb 2010 | | |
| 2 | 17 Dec 2010 | | |
| | 20 Mar 2011 | 13.8 | 0.275 |




| 3 | 17 Mar 2012 | 18.5 | 0.320 |
|---|---|---|---|
|   | 13 Apr 2012 |      |       |
| 5 | 28 Feb 2014 | 23.8 | 0.280 |

Liu et al. (2021) found that both $K_{sh}$ and $n_c$ were lowest immediately after the fire. $K_{sh}$ increased, on average, by approximately
4 mm/h/yr during the first five years of recovery, whereas $n_c$ increased by more than a factor of two after 1 year of recovery
and then remained relatively constant. We focus here on simulating the response to rainfall in the first five years following the
fire where the watershed is likely most vulnerable to extreme responses. To represent the temporal changes in $K_{sh}$ and $n$
documented by Liu et al. (2021) following the fire, we used different values of $K_{sh}$ and $n_c$ for each post-fire year (i.e. post-fire
years 1, 2, 3, and 5) based on the values calibrated by Liu et al. (2021) in post-fire years 1, 2, 3, and 5 (Table. 1). Liu et al.
(2021) were unable to calibrate the necessary K2 parameters in post-fire year 4 so we do not perform any simulations to
constrain flash flood thresholds in that year. Initial soil moisture is set to a volumetric soil-water content of 0.1, following Liu
et al. (2021). Other parameters were also given the same values as the calibrated K2 model, including saturated hydraulic
conductivity of channels (1 mm/hr), net capillary drive of channels (5 mm), hydraulic roughness of hillslopes (0.1 s/(m$^{1/3}$)),
net capillary drive of hillslopes (50 mm), and soil porosity of 0.4. With this model set-up, we simulate the response to each of
the 170 rainstorms for post-fire years 1, 2, 3, and 5.

**3.4 Rainfall intensity-duration thresholds**
Each K2 simulation results in a modeled hydrograph at the watershed outlet. As a first step towards defining a flash flood
threshold, it is necessary to determine, based on the modeled time series of discharge, whether or not a flash flood would have
occurred. We defined the flash flood level as the discharge required to exceed bankfull flow (Sweeney, 1992), which we
assumed was equal to the two-year flood (Leopold et al., 1964). To determine the discharge associated with the two-year flood,
we performed a flood frequency analysis using HEC-SSP v2.2 (Bartles et al., 2019) based on annual maximum records at the
USGS stream gage station (11098000). The discharge associated with the two-year flood at the stream gage station is 15.3
m$^3$/s, with a 95% confidence interval of 12.3-19.2 m$^3$/s (Figure S3). A flash flood threshold by this definition can be viewed
as conservative since it may only indicate the onset of minor flooding as water begins to spill out of the channel. Based on this
definition, we then used two approaches to identify the rainfall ID threshold for flash floods (Figure 2).

The first approach is based on a linear regression analysis that relates peak discharge with different rainfall ID metrics, namely
$I_D^j$ for different values of $j$ and $D$. Using simulations of 170 rainfall-runoff events in each post-fire year, it is possible to
determine a relationship for peak discharge ($Q$) as a function of $I_D^j$. Then, the rainfall ID threshold can be found by determining





the rainfall intensity at which the peak discharge exceeds the bankfull capacity. The simplest quantitative relation is a linear
regression:

$$Q = mI_D^j + k \qquad (2)$$


where $Q$ is the peak discharge (m³/s) of a simulated hydrograph at the outlet, $I_D^j$ denotes rainfall intensity (mm/hr) for the
rainstorm that produced the hydrograph, and $m$ and $k$ denote the slope and y-intercept of the linear regression, respectively.

Considering the channel dimensions and resolution of the DEM used in the K2 model, we selected intensity-discharge ($I_D^j$ -$Q$)
pairs associated with $Q$ greater than 2 m³/s. The parameters in the linear equation (1) with the maximum determination
coefficient ($R^2_{max}$) were estimated using least-squares linear regression in the SciPy Python library for the selected $I_D^j$-$Q$ pairs.
A total of 495 linear regressions were produced for each year because $I_D^j$ can take on 495 different values (5 durations, 99
percentiles) for each rainstorm. For each post-fire year, we then identified the maximum $R^2$ value for each duration as a
function of percentile from 1st to 99th (Figure 3). The rainfall ID threshold for flash flooding in each year was found, for each
duration, from the linear relation associated with the largest $R^2$ (Figure 4).

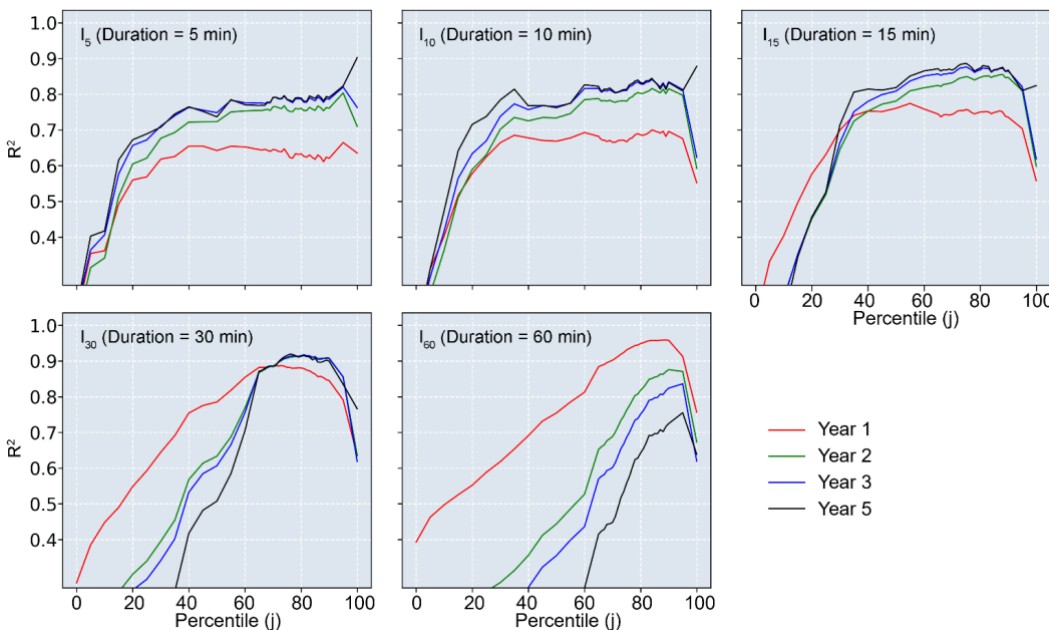


**Figure 3: The determination coefficient ($R^2$) associated with the linear regression between $I_D^j$ and peak discharge in**
**post-fire year 1, 2, 3, and 5. Data used to fit the linear relation is from events with peak discharge greater than 2 m³/s.**

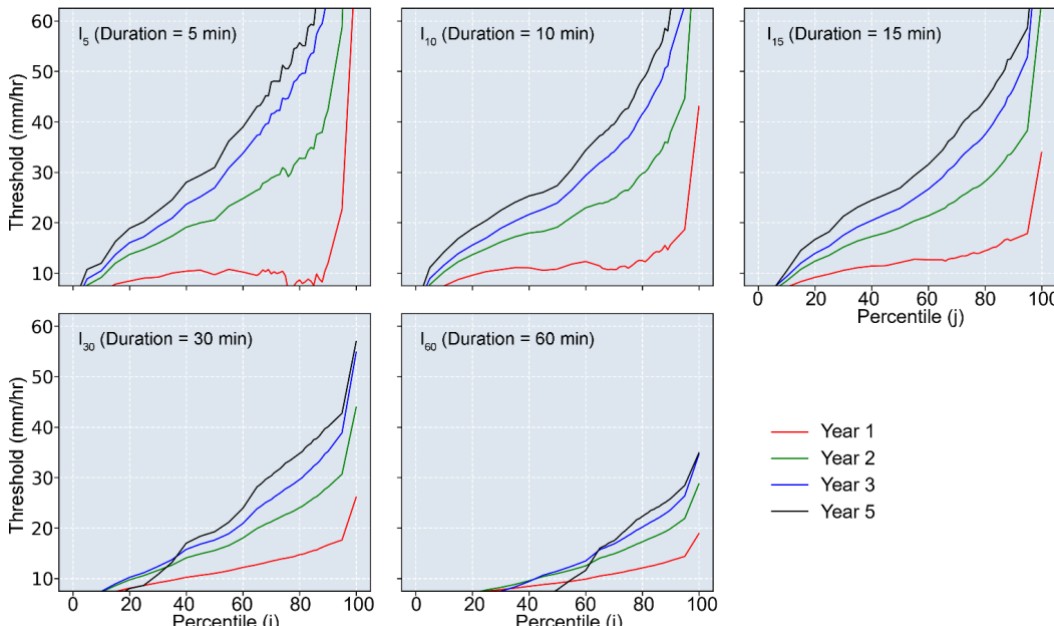

**Fig. 4 The rainfall intensity threshold for flash flood derived from the best linear relation for different durations and percentiles of the most intensive rainfall field in post-fire year 1, 2, 3, and 5.**

The second approach for determining rainfall ID thresholds is based on a receiver operating characteristic (ROC) analysis following Staley et al. (2013). We assess the utility of a potential threshold (e.g. $I_{30}^{50} = 20 mm/hr$), by computing the threat score (TS) associated with using that threshold to define the transition between rainstorms that produce flash floods and those that do not. The TS, as one of the ROC utility functions, measures the fraction of forecast events that were correctly predicted:

$$TS = \frac{TP}{TP + FP + FN}$$  (3)

where TP, FP, and FN denote a true positive, false positive, and false negative, respectively. Flash flood occurrence (true or false) is determined by comparing the peak discharge of each simulated hydrograph with the flash flood level (15.3 m³/s). A TP represents an event where rainfall rates exceed the threshold (e.g. $I_{30}^{50} = 20 mm/hr$), and a flash flood occurred. A FP represents an event where rainfall rates exceed the threshold, but no flash flood occurred. FN events occur when rainfall rates were below the threshold, yet a flash flood occurred. The optimal TS is 1, meaning use of the threshold resulted in no false positives or false negatives.

For a given rainfall intensity metric (e.g. the peak 75th percentile of $I_{30}$, $I_{30}^{75}$, in year 1), we calculated TS for intensities ranging from 0-100 mm/hr at 0.01 mm/hr intervals (Figure 5). We then identified the threshold associated with the maximum TS



(TS$_{max}$). The intensity associated with TS$_{max}$ is the optimal threshold for that rainfall metric (Figure 6). We determined the
optimal threshold associated with each of the 495 rainfall metrics for each post-fire year (1,2,3, and 5) (Figure 7).

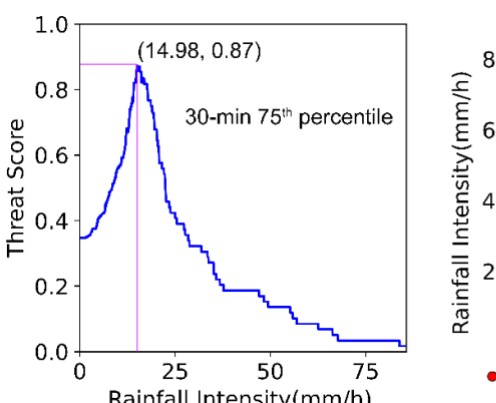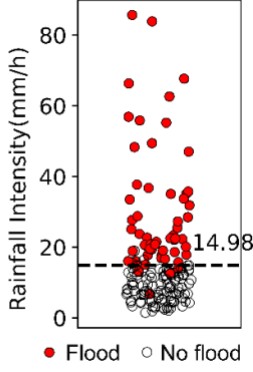


**Fig. 5 Threat score (TS) of the peak 75$^{th}$ percentile of $I_{30}$ in post-fire year 1. (a) Relationship between rainfall intensity**
**and TS; (b) Scatter plots of positive (flood, red circle) and negative (no flood, hollow circle) with the rainfall intensity**
**associated with the maximum TS.**

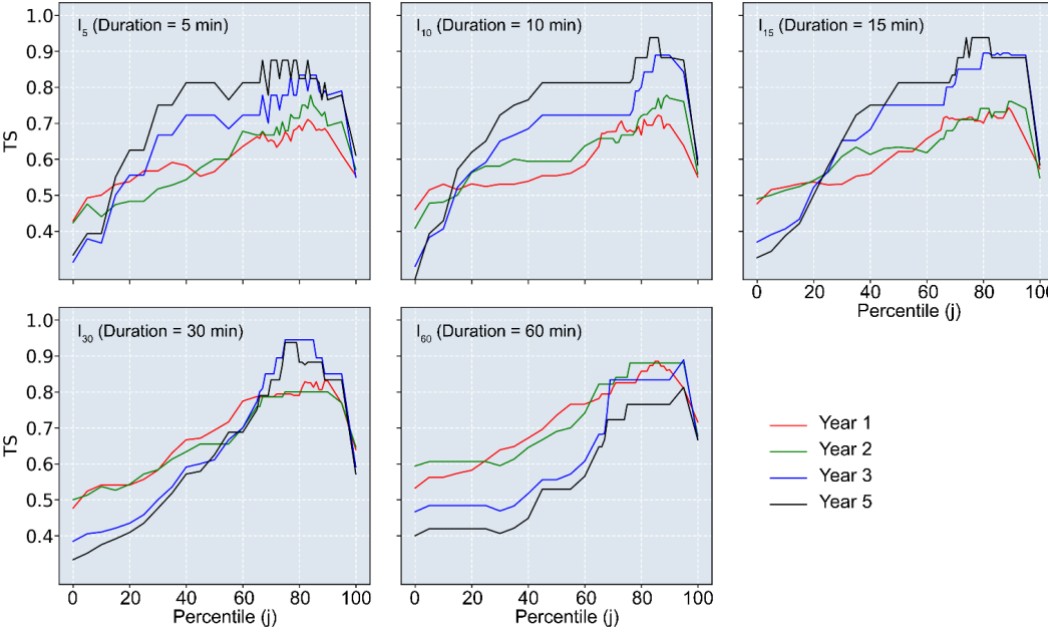


**Fig. 6 The threat scores (TS$_{max}$) associated with flood occurrence and $I_D^j$ in post-fire years 1, 2, 3, and 5. Data used to**
**analyze is from events with peak discharge greater than 2 m$^3$/s.**

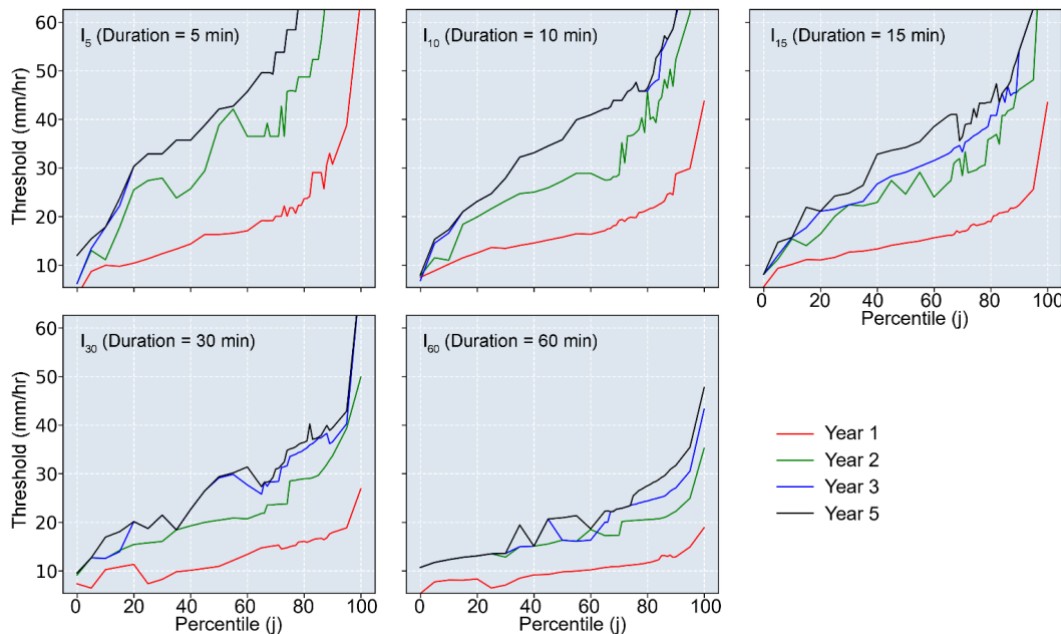


**Fig. 7 The rainfall intensity threshold for flash flood derived from the maximum of TS for different durations and percentiles of the most intensive rainfall field in post-fire years 1, 2, 3, and 5.**

## 4 Results

### 4.1 Optimal summary metrics for defining rainfall ID thresholds

Linear regression analyses suggest that there is a stronger relationship between $I_D^j$ and peak discharge ($Q$) as $j$ increases, with the exception of a rapid dropoff in $R^2$ for j>90 and durations ($D$) greater than 5 minutes (Figure 3). For durations of 5-15 min, $R^2$ were low in the first 20-30 percentiles, then increased to 0.61-0.82 between the 30th-90th percentiles. Whereas the high $R^2$ interval for durations of 30 min and 60 min were with the largest value between 0.92-0.96 between the 60th-90th percentiles in year 1-5. The optimal rainfall threshold for flash floods (based on regressions of $Q$ as a function of $I_D^j$) increased from 13.3 mm/hr of $I_{60}^{89}$ (the 89th percentile of 60 min peak rainfall field) in year 1 to 33.2 mm/hr of $I_{30}^{76}$ (the 76th percentile of 30 min peak rainfall field) in year 5 (Figure 4; Table 2). More generally, averaging rainfall intensity over a duration of 30 minutes and choosing a percentile, $j$, of approximately 75-85 leads to threat scores of approximately 0.8 or greater for all post-fire years. None of the other rainfall summary metrics performed this well across all post-fire years.

**Table. 2 The optimal metrics of rainfall ID and corresponding rainfall thresholds for flash floods in post-fire year 1-5**

| Linear regression | | | Receiver operating characteristic (ROC) |
|---|---|---|---|




| Year | Rainfall metric | Equation | $R^2_{max}$ | Intensity (mm/hr) | Rainfall metric | $TS_{max}$ | Intensity (mm/hr) |
|---|---|---|---|---|---|---|---|
| 1 | $I^{89}_{60}$ | $Q = 10.25 * I^{89}_{60} - 121.27$ | 0.958 | 13.3 | $I^{85}_{60} - I^{86}_{60}$ | 0.89 | 13.1-13.2 |
| 2 | $I^{81}_{30}$ | $Q = 2.38 * I^{81}_{30} - 42.64$ | 0.916 | 24.4 | $I^{76}_{60} - I^{95}_{60}$ | 0.88 | 20.4-25.0 |
| 3 | $I^{81}_{30}$ | $Q = 1.91 * I^{81}_{30} - 41.92$ | 0.917 | 30.0 | $I^{75}_{30} - I^{85}_{30}$ | 0.94 | 33.5-37.3 |
| 5 | $I^{76}_{30}$ | $Q = 2.38 * I^{76}_{30} - 63.70$ | 0.919 | 33.2 | $I^{75}_{30} - I^{79}_{30}$ | 0.94 | 35.1-36.3 |


Note: We denote the peak $j^{th}$ percentile of $I_D$ (rainfall intensity over a duration $D$) as $I^j_D$. For example, $I^{81}_{30}$ is the peak value of
the $81^{st}$ percentile of $I_{30}$ (rainfall intensity over 30-min).

Thresholds derived using the ROC method yielded broadly similar trends. The maximum threat score, $TS_{max}$, generally
increased with j up to a point (approximately j=90) and then began to decrease regardless of the choice of duration ($D$) (Figure
6). The highest threat scores (TS), regardless of post-fire year or duration, were generally associated with the $60^{th}$-$95^{th}$
percentiles. For events in years 1-2, the $TS_{max}$ (0.88-0.89) occurs around $I^{85}_{60}$ (the $85^{th}$ percentile of the peak $I_{60}$ rainfall field);
for events in years 3-5, the $TS_{max}$ (0.94) occurs $I^{75}_{30}$-$I^{79}_{30}$ (the $75^{th}$-$79^{th}$ percentile of the peak $I_{30}$ rainfall field). The optimal
rainfall threshold for flash flood increased from 13.1 mm/hr of $I^{85}_{60}$-$I^{86}_{60}$ (the $85^{th}$-$86^{th}$ percentile of 60 min peak rainfall field)
in year 1 to 36.3 mm/hr of $I^{75}_{30}$-$I^{79}_{30}$ (the $75^{th}$-$79^{th}$ percentile of 30 min peak rainfall field) in year 5 (Table 2; Figure 6). As with
thresholds derived using the linear regression analysis, averaging rainfall intensity over a duration of 30 minutes and choosing
a percentile, $j$, of approximately 75-85 leads to threat scores of approximately 0.8 or greater for all post-fire years. Other
metrics did not perform this well, on average, across all post-fire years.
**4.2 Increases in rainfall intensity thresholds with time since fire**
The rainfall intensity thresholds at each percentile significantly increased from post-fire year 1 to 5 (Figures 4 and 7). However,
the increase from year 1 to 2 is considerably larger than that from year 2 to 3 or from year 3 to year 5. Taking the $I^{75}_{30}$ (the $75^{th}$
percentile of the peak $I_{30}$ rainfall field) as an example due to its strong performance as a threshold for all post-fire years, the
thresholds based on linear regression analyses in year 1, 2, 3, and 5 are 14.0, 22.6, 27.8, and 32.9 mm/hr, respectively; the
ROC-based $I^{75}_{30}$ thresholds in year 1, 2, 3, and 5 are 15.0, 28.5, 33.5, and 35.0 mm/hr, respectively (Figure 7).

We are also able to use the model to assess the individual impacts of temporal changes in $K_{sh}$ and $n_c$ on temporal variations in
the flash flood threshold. If $K_{sh}$ is allowed to vary from year to year (Table 1) and $n_c$ is held fixed at its calibrated value for
year 1, then ROC analysis indicates that the optimal threshold of $I^{75}_{30}$ still increases with time since burning (Figure 8).
However, it increases slower than the case where both $K_{sh}$ and $n_c$ are allowed to vary with time (Figure 8). If $n_c$ is allowed to
vary from year to year (Table 1) and $K_{sh}$ is held fixed at its calibrated value for year 1, then ROC analysis indicates that the
optimal threshold associated with $I_{30}^{75}$ increases from year 1 to year 2 but then stays roughly constant as time increases (Figure
8). Therefore, changes in $K_{sh}$ and $n_c$ both play important roles in determining the degree to which the flash flood threshold
increases from year 1 to year 2, but that further increases in the threshold in years three and five are driven mainly by increases
in $K_{sh}$ as a function of time since burning.

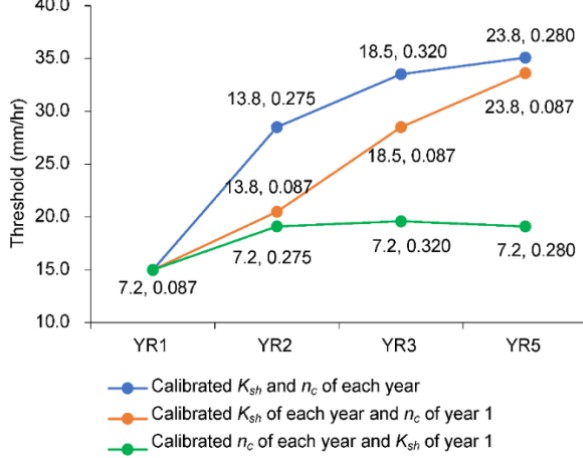

**Figure 8**: **The ROC (receiver operating characteristic) based thresholds for $I_{30}^{75}$ in each year with different model**
**settings. Pairs of $K_{sh}$ (saturated hydraulic conductivity on hillslopes) and $n_c$ (Manning's $n$ in channels) in each model**
**are along with the data points.**
**5 Discussion**
**5.1 Implication of optimal metrics of rainfall intensity for flood warning**
Rain gage records, which provide rainfall intensity data at a single point, are often used to define rainfall ID thresholds in
debris-flow and flash flood studies (e.g. Moody and Martin, 2001; Cannon et al. 2008; Cannon et al. 2011; Guzzetti et al.
2008; Kean et al., 2011; Staley et al., 2013; Raymond et al., 2020; McGuire and Youberg, 2020). Using point source data to
define thresholds for debris flows and flash floods is ideal when rainfall intensity does not vary substantially over the
watershed, an assumption that is most appropriate for watershed areas less than several square kilometers. Radar-derived
rainfall data has the advantage of providing spatially explicit information over an entire watershed at a high-temporal resolution
(e.g. 5 minute). However, one challenge in using radar-derived precipitation to define thresholds is the need to condense
spatially and temporally variable rainfall intensity information down to a single rainfall intensity metric. Regardless of whether
the approach to determining an ID threshold involves fitting empirical relationships (e.g., Moody and Martin, 2001; Cannon



et al., 2008) or using ROC analysis (e.g., Staley et al., 2013), a single metric is required to represent the rainfall intensity for
each duration.

We summarized spatially variable rainfall intensity data over the watershed by computing the peak value of $I_D^j(t)$, the j$^{th}$
percentile of $I_D(t)$ for each rainstorm. We used two different techniques, one based on a linear regression analysis and one
based on ROC analysis (Figure 2), to define thresholds for flash floods in post-fire years 1, 2, 3, and 5. Although the optimal
metrics produced by the two approaches are not identical, they are generally similar in each post-fire year. In particular, high
$R^2$ and $TS_{max}$ values are associated with metrics of the peak 75$^{th}$-85$^{th}$ percentile of rainfall intensity averaged over 30-60 minutes
($I_D^j$ for $75 \leq j \leq 85, D = 30,60$). In other words, a good indicator of the potential for a flash flood is the presence of intense
pulses of rainfall over durations of 30-60 minutes that cover at least 15%-25% of the watershed (Figure 9). This finding
highlights the ability of rainstorms to produce flash floods even if they don't cover the majority of the watershed with intense
rainfall. If rainfall over the majority of the watershed was required to produce flash floods, then we would expect that $I_D^j$ with
j<50 would be a better predictor of flash floods. Previous work has also identified that 30-minute rainfall intensity works well
for predicting flash floods and debris flows (Moody and Martin, 2001; Kean et al., 2011; Staley et al., 2013). The finding that
$I_{30}^j$ and $I_{60}^j$ work best as thresholds when $75 \leq j \leq 85$ could be helpful when issuing flash flood warnings based on radar-
derived precipitation estimates or data from several real-time rain gages within a watershed. Current operational forecast
models such as the High Resolution Rapid Refresh model have a horizontal resolution of 3km and minimum temporal
resolution of 15 minutes (Benjamin et al. 2016; NOAA 2021a), such that it is feasible to use $I_{30}^j$ and $I_{60}^j$ in an operational
forecast setting. Where sufficient operational NEXRAD weather radar coverage is present, radar-derived precipitation
estimates such as the MRMS (Zhang et al. 2016) can provide near-real-time precipitation estimates at 1 km and as fine as 15
min temporal resolution (NOAA 2021b). In the case of poor radar coverage, gap-filling radars may be temporarily deployed
or installed (e.g., Jorgensen et al. 2011; Cifelli et al. 2018) to provide information necessary for accurate precipitation estimates.
While the magnitude of rainfall thresholds estimated here may only work for similar, recently burned watersheds within the
San Gabriel Mountains, the use of metrics such as $I_{30}^{75}$ as a reliable predictor of post-fire flash floods may be more general.
Further testing is needed in watersheds with different watershed size, topographic characteristics, landscape, and burn severity
patterns.


**Figure 9: Snapshots of the spatial patterns of $I_{30}^{75}$ of 34 unique storms. The peak j[th] percentile of $I_D$ (rainfall intensity over a duration $D$) is denoted as $I_D^j$. $I_{30}^{75}$ is the peak value of the 75st percentile of $I_{30}$ (rainfall intensity over 30-min). Red contours delineate the pixels with rainfall intensities larger than $I_{30}^{75}$ of each storm.**

Several limitations are present in this work. First, we assess a small number of storm events (34) in the area as we are limited by the length of radar and gage records as well as and the number of events that impact the indicator rain gages. However, the advantage of using observed storms rather than using a rainfall generator (e.g., Zhao et al., 2019; Evin et al., 2018) is that our results represent spatial and temporal precipitation patterns that are physically realistic. Second, the challenges of radar observations and application of Z-R relationships to convert reflectivity to precipitation also presents challenges in accurately representing precipitation values. This can be addressed in future work through studies to constrain Z-R relationships for


storms producing intense rainfall in this region and through the deployment or installation of high-resolution gap-filling radars
(e.g., Johnson et al. 2019).

**5.2 Increasing rainfall intensity thresholds with time since fire**

In this study we employed the K2 model calibrated by Liu et al. (2021) to parameterize hydrologic changes affecting Hortonian
overland flow within a five-year period following fire. Hillslope saturated hydraulic conductivity ($K_{sh}$ = 7.2 mm/hr) and
hydraulic roughness in channels ($n_c$ = 0.087 s/m$^{1/3}$) were lowest immediately after fire (Table 1), resulting in high runoff
coefficients and low rainfall thresholds in post-fire year 1. In later years, with $K_{sh}$ and $n_c$ gradually increasing (Table 1), more
rainfall infiltrated into soil and there was increased attenuation of flood peaks. Simulations indicate that the number of flash-
flood-producing rainstorms decreased from 59 in year 1 to 25, 18, and 16 in years 2, 3, and 5, respectively. Runoff coefficients
and peak discharge of simulated hydrographs also decreased with time since fire (Figure 10). Given the same precipitation
ensemble, the likelihood of flash floods significantly decreased with time. The peak discharge produced by the highest intensity
rainfall event with $I_{60}^{75}$ of 51.8 mm/hr was 554.0 m$^3$/s in the first year after the fire, which is three times greater than the peak
discharges of 157.5 m$^3$/s in year 3 and 161.2 m$^3$/s in year 5 produced by the same rainstorm. From a flood hazard perspective,
the downstream area may be exposed to a 1000-year flood under the recently burned condition (less than one year since the
fire), whereas the discharge produced in years three and five would amount to roughly a 30- to 40-year flood (Figure S3).

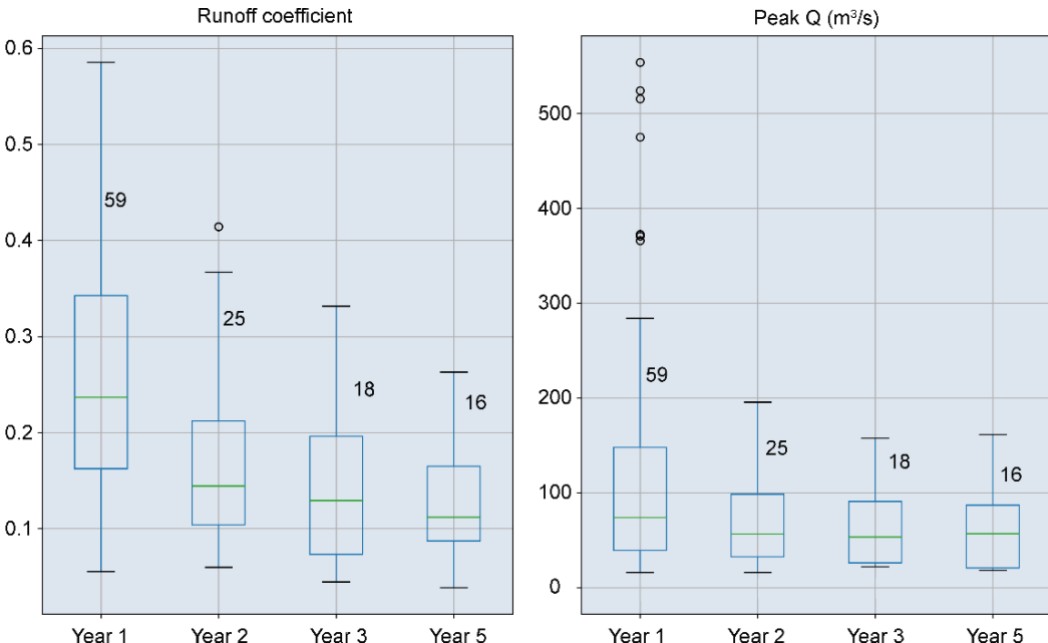


**Figure 10: Box plots showing the runoff coefficient and peak discharge of flash floods in post-fire year 1, 2, 3, and 5.**
**The numbers of flash floods in each year are displayed next to the box.**





We were also able to perform numerical experiments to quantify the relative importance of temporal changes in $K_{sh}$ and $n_c$ on
temporal variations in the flash flood threshold (Figure 8). Results suggest that changes in vegetation and grain roughness,
which are likely to influence $n_c$, throughout the recovery process are less important for determining flash flood potential in our
study area relative to changes to saturated hydraulic conductivity on hillslopes. It is worth noting that temporal changes in
other model parameters (e.g., hydraulic roughness on hillslopes, capillary drive) may play more of a role in driving changes
in post-fire flash flood thresholds in other settings. In this study, however, we focus on changes in $K_{sh}$ and $n_c$ because Liu et
al. (2021) were able to detect temporal changes in $n_c$ and $K_{sh}$ through time and unable to detect similar temporal changes in
other hydrologic parameters (e.g., hydraulic roughness on hillslopes, capillary drive) due to their relatively minor influence on
runoff in the study watershed.

In this study, the optimal flash flood thresholds increased from $I_{30}^{75}$ =14.0-15.0 mm/hr in post-fire year 1, to 22.6-28.5 mm/hr
in year 2, and 22.9-35.1 mm/hr in post-fire year 5 (Figure 4 and 7; Table 2). In the San Gabriel Mountains and nearby San
Bernardino and San Jacinto Mountains, Cannon et al. (2008) estimated rainfall thresholds of $I_{30}$=9.5 mm/hr and for flash floods
and debris flows in the first winter rainy season following fire. They found that the thresholds for flash floods and debris flows
increased to $I_{30}$=19.8 mm/hr in post-fire year 2. The thresholds that we infer from hydrological modeling are greater than those
reported by Cannon et al. (2008), which may be partly due to differences in (1) data and methods used and (2) the size of the
studied watersheds. Our results are driven by a hydrologic model, forced with a radar precipitation ensemble that consists of
170 rainstorms that contain a variety of storm types that impact southern California. The occurrence of a flash flood is based
on exceedance of the maximum channel capacity and we summarize temporal changes in the rainfall ID threshold using $I_{30}^{75}$
since we find this to be a reliable metric for all post-fire years included in this study. In contrast, Cannon et al. (2008)
established rainfall ID relations by using observations of rainstorms and hydrological response in the two years following fire
in 87 small watersheds (0.2-4.6 km$^2$). They base their thresholds on rainfall characteristics that produced either flash floods or
debris flows whereas we focus solely on flash floods. In their dataset, flash floods and debris flows were identified by
investigating flood and debris flow deposits at the outlet of those small watersheds in the field. Despite differences in the
magnitude of the thresholds, the increase in the threshold from post-fire year 1 to year 2 in both studies are quite close. This
agreement provides support for the use of simulation-based approaches to inform temporal shifts in rainfall ID thresholds.

During the recovery process, increasing thresholds for flash floods and debris flows have also been identified in other areas at
different scales by either observation- or simulation-based studies, such as hillslopes in the Colorado Front Range (Ebel, 2020)
and small watersheds in Australia (Noske et al., 2016). The consistent increase in rainfall ID thresholds with time since fire in
different geographic and ecological zones implies that hydraulic and hydrologic models may be useful tools for exploring how
transient effects of fire translate into changes in water-related hazards. Particularly when historic data is limited and traditional
empirical methods are impractical for defining thresholds, the role of hydraulic and hydrological models becomes more
important.





## 6 Conclusions

We used 250 m, 5-minute radar-derived precipitation estimates over a 41.5 km$^2$ watershed in combination with a calibrated hydrological model to estimate the rainfall intensity thresholds for post-fire flash floods as a function of time since burning. The optimal threshold for predicting the occurrence of a flash flood in our study areas is the 75th-85th percentile of peak rainfall intensity averaged over 30-60 minutes, i.e., $I_{30}^{75}$-$I_{30}^{85}$. In other words, a flash flood tends to be produced when rainfall intensity over 15%-25% of the watershed area exceeds a critical value. A threshold based on $I_{30}^{75}$ performs consistently well for post-fire years 1, 2, 3, and 5, although the magnitude of the threshold increases with time since burning. For the watershed studied, the $I_{30}^{75}$ threshold increases from 14.0-15.0 mm/hr for year 1 to 22.6-28.5 mm/hr, 27.8-33.5 mm/hr, and 32.9-35.1 mm/hr, for years 2, 3, and 5 respectively. Increases in the threshold value of $I_{30}^{75}$ can be primarily attributed to increases in $K_{sh}$ rather than $n_c$ during the hydrological recovery process. The increase in the magnitude of the threshold from year 1 to year 2 is consistent with previous observations from nearby areas in southern California. Results provide a methodology for using radar-derived precipitation estimates and hydrological modeling to estimate flash flood thresholds for improved warning and mitigation of post-fire hydrologic hazards. Thresholds developed through these methods can then be built into operational tools that use incoming radar data to evaluate flash flood hazard in near-real time or precipitation forecasts to evaluate potential for flash flood hazard in burned watersheds.

## Author contributions

TL and LM conceived the study. TL, LM, NO and FC contributed to the development and design of the methodology. TL analysed and prepared the manuscript with review and analysis contributions from LM, NO and FC.

## Competing interests

The authors declare that they have no conflict of interest.

## Acknowledgments:

Haiyan Wei, Carl L. Unkrich, and David C. Goodrich, who are from the KINEROS2 development group in the USDA ARS Southwest Watershed Research Center in Tucson, helped with the setting up of the KINEROS2 model and ingestion of the RADAR precipitation data into the model. We are thankful for their great help.





**Financial support**

This work was supported by the National Oceanic and Atmospheric Administration (NOAA) Collaborative Science, Technology, and Applied Research (CSTAR) Program under grant NA19NWS4680004 and by the National Integrated Drought Information System (NIDIS) through Task Order 1332KP20FNRMT0012.

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
