# Peer review of "Temporal changes in rainfall intensity-duration thresholds for post- 2 wildfire flash floods in Southern California"

_Natural Hazards and Earth System Sciences, 2021_

## Referee Comment (RC2)

Using a dynamic hydrologic model with varying parameters with the recovery of a burned watershed in California and radar rainfall estimates, the authors investigated changes in rainfall intensity-duration thresholds inducing flash floods in the watershed as it recovers from fire. The study is of practical value for informing post-fired flash floods. My comments are as follows.

1. The title does not accurately summarize what has been done in the study as there is little information about how the spatiotemporal distributions of rainfall affect the intensity-duration thresholds inducing flash floods or their changes with recovery.

2. Given the small sample of rainfall events (34 in total), should sampling uncertainty (thus robustness) of the estimated intensity-duration thresholds be evaluated. One simple method might be to bootstrap these events or values of I(D, j), with which to obtain a set of plausible estimates of intensity-duration thresholds and evaluate robustness of the reported results.

3. There are several prescribed quantities, such as a value of 15mm/h for extracting moderate-to-high rainfall events and a value of 2m^3/s for defining effective discharge. I did not see discussions about the rationale of these choices, nor their potential influences on the identified intensity-duration thresholds inducing flash floods.

4. The whole third paragraph on page 6 is spent on describing the dominating spatial patterns of rainfall events in the study watershed. Nevertheless, they seem to be forgotten after that. How these spatial patterns affect the identified intensity-duration thresholds and their changes? See my first comment.

5. Lines 174-176, given such many ignored factors that may influence the reliability of derived radar rainfall estimates, how "the realistic spatial and temporal patterns of rainfall" can be guaranteed?

6. Line 179, as the used Z-R relationships are not calibrated for the study watershed, the derived rainfall events are not "realistic storms". Replace "realistic" with "plausible".

7. Page 8, a more concrete interpretation to I(D, j) can substantially improve readability. Possibly, adding somewhere "I(D, j) indicates that 100(1-j)% of the watershed experiences rainfall of duration D with intensity  of I or larger."

8. Line 383, the statement is not convincing.

9. Lines 394-397, inaccurate statements. Are all existing rainfall generators unable to produce physically realistic rainfall fields? I think this question has been addressed in many published studies.

---

## Author Comment (AC1)

We appreciate Referee #1's careful review and insightful suggestions. We truly believe that the changes suggested by Referee #1 will enhance the quality of the manuscript.

Comments to the Author:

Wildfire is an important interference factor in the forest ecosystem. The occurrence and development of mountain torrents after a disaster has a complex mechanism, which is affected by many factors. Based on the precipitation obtained by hydrologic model and radar technology, estimate the identification threshold of flash floods after fire in the Southern California basin of the United States, and analyze the change characteristics of the threshold over time, providing reference for forest disaster prevention, disaster prevention and mitigation planning, and climate change adaptation in accordance. The article has a certain degree of logic and content, and the overall research is of strong practical significance. It can be included in this journal, but there are still some problems in the article, and it is recommended to modify it accordingly.

R: We appreciate the reviewer's insightful comments.

**Introduction**

This part of the content is the current research status and progress of flash floods after the disaster, and points out that it is impossible to use a single indicator to characterize the threshold of flash floods. However, it seldom describes the innovation and research significance of the article, and it is recommended to add relevant content and modify it.

R: Combining the calibrated model with radar rainfall data, we demonstrate the optimal rainfall ID metrics as flash flood thresholds and analyze the temporal changes of the optimal rainfall ID thresholds through the first five years of recovery following the 2009 Station Fire in Southern California, where is densely populated, fire-prone region.

On one hand, it provides insight into the trend of the flash flood thresholds following a fire in a chaparral-dominated environment similar to the southern California. On the other hand, the optimal rainfall ID metrics as flash thresholds identified in this study could be helpful when issuing flash flood warnings based on radar-derived precipitation estimates or data from several real-time rain gages within a watershed.

We added these content in Introduction.

*"Results provide insight into the magnitude of temporal changes in flash flood thresholds in the densely populated, fire-prone region of southern California. Findings also provide guidance for the magnitude of change expected in rainfall ID thresholds for flash floods during the post-fire recovery period in chaparral-dominated environments similar to southern California. More generally, results support the development of early warning systems for flash floods by identifying specific metrics that can be computed using spatially variable rainfall intensity estimates to assess the potential for flash flooding. The optimal rainfall ID metrics identified in this study could be helpful when issuing flash flood warnings based on radar-derived precipitation estimates or data from several real-time rain gages within a watershed."*

**Research area**

â The format of the title of the article is not uniform. For example, "Figure 1, Figure 2 and Fig. 4" suggest to unify the format and modify it according to the requirements of the journal.

R: The format of all figure title is modified according to the journal's requirements.

â The article is rigorous in description and rich in content, but there are some detailed problems, such as the inconsistent citation format, (Staley et al., 2014). and (Oakley et al. 2018a). It is recommended to organize and modify according to the requirements of this journal.

R: We modify all citations in the format of the journal.

**Data and methods**

â The article estimates the value of precipitation based on hydrological models and radar technology. It is recommended to explain whether the result has passed relevant verification.

R: The precipitation is estimated from radar observations, which are archived for the two operational NWS Next-Generation Weather Radar installations (NEXRAD; NOAA 1991) that cover the study area, KSOX, (Santa Ana), and KVTX (Ventura). The reflectivity (R) data is then converted to precipitation by using five well-known and previously published Z-R relationships were applied to the gridded reflectivity values. Supplement S3 lists the different Z-R relationships applied here and the general conditions for which they are suitable.

â¡The article points out that after wildfires, mountain torrents erupt, saturated hydraulic conductivity on hillsides and hydraulic roughness in river channels play a decisive role, but whether it is necessary to consider the situation of burned sites, channel conditions, underlying surface conditions, and development history. Consider only the rainfall intensity, one of the material sources of the flood.

R: Good point. Many factors can affect the occurrence and magnitudes of post-fire flash floods. However, we employ a well-calibrated hydrological model for the upper Arroyo Seco (Liu et al., 2021), in which many factors have been considered, to explore changes in rainfall ID thresholds following disturbance in the study watershed. For other watersheds, it is necessary to consider all important factors when using a hydrological model to infer rainfall ID thresholds.

â¢The article is based on the K2 hydrologic model to simulate rainfall division, surface watershed and flood path. This model has relative advantages over other hydrologic models. In this study area, this hydrologic model has fewer verification links and overall adaptability evaluation descriptions. It is recommended to consider adding related information.

R: KINEROS2 (K2) is an event-scale, distributed-parameter, process-based watershed model, which has been used extensively for rainfall-runoff processes in semi-arid and arid watersheds (Smith et al., 1995; Goodrich et al., 2012). Specifically, this model has been well calibrated using rain gage data in combination with the USGS stream gage installed at the outlet of the upper Arroyo Seco watershed during different stages of the post-fire recovery process (Liu et al., 2021).

â£This paragraph is rich in content and complete in structure, but there are still some details. It is recommended to organize and modify it. For example, "It is written in the text (Table S1) but this table does not appear. The format of the table is recommended to be modified according to the journal's requirements. The three-line format shall prevail , So that readers can read and understand".

R: Table S1 is included in Supplements (S2) according to the journal's requirements.

**Results**

There are some detailed problems in the content of this paragraph. For example, "the horizontal and vertical coordinate units of some charts are not marked" are suggested to be added so that readers can understand the relevant content.

R: Y axis labels of Figure 10 have been added.

**Discussion**

This paragraph points out the significance of the optimal measure of rainfall intensity after a fire for flood warning and the increase in rainfall intensity threshold over time.

â It is recommended to increase the comparison between the research of this article and the research conclusions of other scholars, as well as the existing shortcomings.

R: Good suggestion. Comparisons can be found in Discussion section 5.1.

*"Previous work has also identified that 30-minute rainfall intensity works well for predicting flash floods and debris flows (Moody and Martin, 2001; Kean et al., 2011; Staley et al., 2013). The finding that $I\_15^{\wedge}j$, $I\_30^{\wedge}j$ and $I\_60^{\wedge}j$ work best as thresholds when $75{\leq}j{\leq}85$ could be helpful when issuing flash flood warnings based on radar-derived precipitation estimates or data from several real-time rain gages within a watershed. Current operational forecast models such as the High Resolution Rapid Refresh model have a horizontal resolution of 3km and minimum temporal resolution of 15 minutes (Benjamin et al., 2016; NOAA 2021a), such that it is feasible to use either $I\_15^{\wedge}j$, $I\_30^{\wedge}j$or $I\_60^{\wedge}j$ in an operational forecast setting. Where sufficient operational NEXRAD weather radar coverage is present, radar-derived precipitation estimates such as the MRMS (Zhang et al., 2016) can provide near-real-time precipitation estimates at 1 km and as fine as 15 min temporal resolution (NOAA 2021b). In the case of poor radar coverage, gap-filling radars may be temporarily deployed or installed (e.g., Jorgensen et al., 2011; Cifelli et al., 2018) to provide information necessary for accurate precipitation estimates. While the magnitude of rainfall thresholds estimated here may only work for similar, recently burned watersheds within the San Gabriel Mountains, this work provides a general methodology for exploring reliable predictors of post-fire flash floods for other watersheds and settings. Further testing is needed in watersheds with different watershed size, topographic characteristics, landscape, and burn severity patterns."*

â¡There are some detailed problems, it is recommended to sort out and modify them, such as "It is written in the text (Figure S3) but this picture does not appear".

R: Figure S3 is included in the Supplement.

â¢The title of 4.2 "Increases in rainfall intensity thresholds with time since fire "is similar to the title of 5.2 "Increasing rainfall intensity thresholds with time since fire". It is recommended to modify it for readers to understand.

R: Good point. We changed the title of 5.2 into,

*"**The role of hydrological models in rainfall intensity threshold estimation**."*

**Conclusion**

It is recommended to classify and elaborate the main research conclusions of the article for readers to understand.

R: We have modified Conclusions section as follows,

*"We used 250 m, 5-minute radar-derived precipitation estimates over a 41.5 km2 watershed in combination with a calibrated hydrological model to estimate rainfall intensity-duration thresholds for post-fire flash floods as a function of time since burning. The main outcomes of this study are 1) identification of optimal radar-derived rainfall metrics for post-fire flash flood prediction in southern California, 2) estimates of temporal changes in rainfall ID thresholds for flash floods following disturbance in a chapparal-dominated ecosystem, and 3) a methodology for using a hydrological model to assess changes in post-fire flash flood thresholds.*

*Results indicate that thresholds based on the 75th-85th percentile of peak rainfall intensity averaged over 15-60 minutes perform best at predicting the occurrence of a flash flood in our study area. In other words, a flash flood tends to be produced when rainfall intensity over 15%-25% of the watershed area exceeds a critical value. A threshold based on $I\_30^{75}$ performs consistently well for post-fire years 1, 2, 3, and 5, although the magnitude of the threshold increases with time since burning. For the watershed studied, the $I\_30^{75}$ threshold increases from 16.0-16.8 mm/hr for year 1 to 23.2-26.9 mm/hr, 26.9-32.6 mm/hr, and 27.6-34.5 mm/hr, for years 2, 3, and 5 respectively. Increases in the threshold value of $I\_30^{75}$ can be primarily attributed to increases in Ksh rather than nc during the hydrological recovery process. The increase in the magnitude of the threshold from year 1 to year 2 is consistent with previous observations from nearby areas in southern California. Results provide a methodology for using radar-derived precipitation estimates and hydrological modeling to estimate flash flood thresholds for improved warning and mitigation of post-fire hydrologic hazards. Thresholds developed through these methods can then be built into operational tools that use incoming radar data to evaluate flash flood hazard in near-real time or precipitation forecasts to evaluate potential for flash flood hazard in burned watersheds."*

**References**

â Some cited documents do not have a year and are unified in accordance with journal requirements.

R: The references list is updated.

â¡The format of URL addition is not uniform, for example, line 485 is from https://journals.ametsoc.org/view/journals/mwre/144/4/mwr-d-15-0242.1.xml. Others are added directly.

R: This reference (line 482-485) is updated.

It is recommended to make unified rectification in accordance with the requirements of the journal.
â¢The font size in the text are not uniform. For example, "35 of line 491, 31 of line 518, and 5 of line 556 are bold fonts, but other references do not have this format, and it is recommended to be unified".

R: The font size and style are updated.

---

## Author Comment (AC2)

We appreciate Referee #2's careful review and insightful suggestions. The followings are our response to comments and revision.

Using a dynamic hydrologic model with varying parameters with the recovery of a burned watershed in California and radar rainfall estimates, the authors investigated changes in rainfall intensity-duration thresholds inducing flash floods in the watershed as it recovers from fire. The study is of practical value for informing post-fired flash floods. My comments are as follows.

1. The title does not accurately summarize what has been done in the study as there is little information about how the spatiotemporal distributions of rainfall affect the intensity-duration thresholds inducing flash floods or their changes with recovery.

R: We changed the title to "Temporal changes in rainfall intensity-duration thresholds for post-wildfire flash floods in Southern California." The spatial analysis was an initial intent of the study, but was not as interesting as expected and therefore not a focus of this paper.

2. Given the small sample of rainfall events (34 in total), should sampling uncertainty (thus robustness) of the estimated intensity-duration thresholds be evaluated. One simple method might be to bootstrap these events or values of I(D, j), with which to obtain a set of plausible estimates of intensity-duration thresholds and evaluate robustness of the reported results.

R: Good suggestion! We bootstrapped original samples obtaining 50 replications. With the resampled events, we estimated 95% confidence intervals for rainfall intensity-duration thresholds (Fig.3-4 and Fig 6-7).

3. There are several prescribed quantities, such as a value of 15mm/h for extracting moderate-to-high rainfall events and a value of 2m^3/s for defining effective discharge. I did not see discussions about the rationale of these choices, nor their potential influences on the identified intensity-duration thresholds inducing flash floods.

R: The 15 mm/h threshold was chosen so that we could extract storms that have the potential to generate floods. This threshold provided a balance between having too many rainfall events to process and having a sufficient number of storms to perform the model and ID threshold analyses. We explained this value in Section 3.1 as follows,

*"This threshold generally corresponds with a 1-year average recurrence interval storm event in the study area (NOAA Atlas 14). This value falls between the California-Nevada River Forecast Center's flash flood guidance for unburned areas in the region (~22-25 mm/h; CNRFC 2021) and regional thresholds for post-wildfire debris flows in this region at a point (12.7 mm/h, Cannon et al. 2008; Staley et al. 2013). This threshold allows us to focus on storms that have a high potential to generate floods, while keeping the number of storms to a manageable level for data processing."*

Low flow is not our focus in this study. Flow depths associated with Q of less than 2 $m^3$/s are very small and any potential impacts from such flow would be negligible.

4. The whole third paragraph on page 6 is spent on describing the dominating spatial patterns of rainfall events in the study watershed. Nevertheless, they seem to be forgotten after that. How these spatial patterns affect the identified intensity-duration thresholds and their changes? See my first comment.

R: Spatial patterns associated with different storm types and their effects on rainfall thresholds were an initial interest of ours, but the results show thresholds are similar across different storm types. For example the I(15, 80) threshold for year 1 for all storm types is roughly 20 mm/hr. Details are attached in Figure S4-5. However, we do still think that it is beneficial to describe the different rainfall patterns to illustrate that our modeling analyses are forced with rainfall data that reflects the different types of systems that impact this area.

5. Lines 174-176, given such many ignored factors that may influence the reliability of derived radar rainfall estimates, how "the realistic spatial and temporal patterns of rainfall" can be guaranteed?

R: While the rainfall intensities at each grid cell and time step will vary somewhat across Z-R relationships, the spatial and temporal characteristics of the storm will be retained. As the radar data are observed rather than simulated, the spatial and temporal patterns offer the best representation of "reality" available over the area. The comment on radar measurement uncertainty is meant to provide general guidance for things to consider when using radar data, but this area exhibits quality radar coverage, as noted now in the first line of section 3.1.

*"Weather radar coverage is adequate for estimating rainfall over the study area (NOAA 2021), and radars have been operational since the mid-1990s. This allows us to utilize observed data to capture temporal and spatial characteristics of storms impacting the study area, a region of complex terrain."*

6. Line 179, as the used Z-R relationships are not calibrated for the study watershed, the derived rainfall events are not "realistic storms". Replace "realistic" with "plausible".

R: Modified.

7. Page 8, a more concrete interpretation to I(D, j) can substantially improve readability. Possibly, adding somewhere "I(D, j) indicates that 100(1-j)% of the watershed experiences rainfall of duration D with intensity of I or larger."

R: We agree, thank you for this suggestion. We added this interpretation in Section 3.2 as follows:

*"A threshold defined by $I_D^j$ would denote a threshold where (100-j)% of the watershed experiences rainfall of duration D with an intensity of I or greater."*

We also described the j[th] percentile in the Discussion and Conclusion sections,

*"In other words, a good indicator of the potential for a flash flood is the presence of intense pulses of rainfall over durations of 15-60 minutes that cover at least 15%-25% of the watershed (Figure 9)."*

*"The optimal threshold for predicting the occurrence of a flash flood in our study areas is the 75[th]-85[th] percentile of peak rainfall intensity averaged over 15-60 minutes, i.e., $I_{30}^{75}$-$I_{30}^{85}$. In other words, a flash flood tends to be produced when rainfall intensity over 15%-25% of the watershed area exceeds a critical value."*

8. Line 383, the statement is not convincing

R: It has been rephrased as, "While the magnitude of rainfall thresholds estimated here may only work for similar, recently burned watersheds within the San Gabriel Mountains, this work provides a general

methodology for exploring a reliable predictors of post-fire flash floods for other watersheds and settings."

9. Lines 394-397, inaccurate statements. Are all existing rainfall generators unable to produce physically realistic rainfall fields? I think this question has been addressed in many published studies.

R: We agree that this statement is misleading and needs revision. We have updated the text in the last paragraph of section 5.1 to increase clarity.

*"Several limitations are present in this work. First, we assess a small number of storm events (34) in the area as we are limited by the length of radar and gage records as well as and the number of events that impact the indicator rain gages, though applying the five Z-R relationships provides us with 170 rainfall realizations to assess. We prefer the use of observed rainfall data (radar and gauges) over simulated products, such as output from a rainfall generator (e.g., Zhao et al., 2019; Evin et al., 2018), as the radar is able to capture the spatial and temporal patterns of rainfall intensity in the study area's complex terrain. Though rainfall generators have advanced to represent some synoptic-to-mesoscale features, such as frontal and convective precipitation (e.g., Zhao et al. 2019), they are fundamentally designed to represent statistical characteristics of rainfall in places with limited observations (Wilks and Wilby 1999) and cannot be relied upon to replicate small scale storm characteristics in complex terrain (e.g., Camera et al. 2016). Future work could compare results from this hydrologic modeling experiment with observed versus simulated rainfall. Second, the challenges of Z-R relationships to convert reflectivity to precipitation also presents challenges in accurately representing precipitation values. This can be addressed in future work through studies to constrain Z-R relationships for storms producing intense rainfall in this region and through the deployment or installation of high-resolution gap-filling radars (e.g., Johnson et al., 2019)."*

Additional modifications were made in the first paragraph of section 3.1 that help explain our preference for observed over simulated data.

*"Weather radar coverage is adequate for estimating rainfall over the study area (NOAA 2021), and radars have been operational since the mid-1990s. This allows us to utilize observed data to capture temporal and spatial characteristics of storms impacting the study area, a region of complex terrain."*